# The Impact of the Need for Language Assistance Services on the Use of Regional Anesthesia, Postoperative Pain Scores and Opioid Administration in Surgical Oncology Patients

**DOI:** 10.3390/jpm13030481

**Published:** 2023-03-07

**Authors:** Ravish Kapoor, Pascal Owusu-Agyemang, Lei Feng, Juan P. Cata

**Affiliations:** 1Department of Anesthesiology & Perioperative Medicine, The University of Texas MD Anderson Cancer Center, Houston, TX 77030, USA; 2Department of Biostatistics, The University of Texas MD Anderson Cancer Center, Houston, TX 77030, USA

**Keywords:** pain assessment, oncologic surgery, language barriers, limited English proficiency, language assistance services

## Abstract

Language barriers can negatively impact the quality of healthcare. In surgical patients, limited English proficiency (LEP) can lead to disparities in acute postoperative pain management. Interpreters are often used for communication with LEP patients to help alleviate these disparities. We aimed to investigate the impact of the need for language assistance services (LAS) in acute postoperative pain management in patients undergoing oncologic surgery. We retrospectively collected data on adult patients undergoing open abdominal oncologic surgery between March 2016 and August 2021. The need for LAS, patient demographics, treatment and clinical outcomes were obtained from the patient’s electronic medical record. The primary endpoint was pain intensity, while secondary endpoints included opioid use in PACU and regional anesthesia. Post-matching analysis (n = 590) demonstrated no significant difference in preoperative variables between patients needing LAS and those not needing LAS. The rate of regional use was slightly lower but not statistically significant in patients needing LAS. Patients needing LAS had significantly lower opioid consumption and reported lower pain intensity in PACU than subjects not requiring translation. In this study, LAS may have aided in the patient decision process regarding the acceptance of regional anesthesia. Although the need for LAS was associated with statistically significant lower pain intensity scores and a corresponding lesser opioid use than no LAS, the margin of differences, especially in pain intensity scores, may not be clinically significant. This may suggest that LAS allowed for better patient-provider communication and appropriate pain management.

## 1. Introduction

In the United States, 8.6% of the population has limited English proficiency (LEP) and speaks English less than “very well” [1]. Language barriers have been shown to negatively impact the ability to deliver and receive quality healthcare [2]. In a recent study, Spanish-speaking patients with LEP were nearly three times more likely not to receive access to usual healthcare compared to English-speaking patients [3]. Furthermore, patients with LEP themselves have perceived poorer patient-physician interactions than those who primarily speak English [4].

Pain is a multidimensional experience in which language proficiency is an important component of pain communication [5]. Aspects of pain such as location, timing, intensity, type, radiation, and alleviating/exacerbating factors must all be considered in effective pain management. Ruppen et al. reported that communication was very difficult in 6–7% of their chronic pain clinic patients who did not speak English or one of the four primary languages spoken in Switzerland [6]. This lack of communication can lead to misdiagnosis and mistreatment of pain. In surgical patients, LEP can lead to disparities in acute postoperative pain management [7]. 

Interpreters are often used for communication with LEP patients to help alleviate these disparities and have been perceived by patients to improve their pain assessment and treatment [8]. However, a lack of or a delay of access to interpreters in high acuity areas such as the postoperative anesthesia care unit (PACU) can present a challenge to patients with LEP who have acute or chronic postsurgical pain. In a recent study of trauma patients, patients had significantly fewer pain assessments per day than patients fluent in English. Interestingly, after adjusting for the frequency of pain assessments, patients proficient in English had significantly higher pain scores than LEP [9]. This is in contrast to Koleck et al., who reported that Spanish or Southeast Asian patients with LEP had an increased chance of reporting any pain compared to English-speaking patients [10].

In this study, we aimed to investigate the impact of the need for language assistance services (LAS) in acute postoperative pain management for patients undergoing oncologic surgery. We hypothesized that patients with LEP would report significantly different pain scores compared to non-LEP subjects, which would be associated with differences in opioid use. 

## 2. Materials and Methods

Following Institutional Review Board approval (IRB#2021-0738), we retrospectively collected data for adult patients who underwent open abdominal oncological surgery between March 1, 2016 and August 1, 2021. We excluded patients whom a) had non-abdominal surgery, laparoscopic surgery, emergency surgery, outpatient surgery, b) were from surgical services that did not utilize regional analgesia, c) were classified as American Society of Anesthesiologists Physical Status > 3 and d) had abdominal plus additional surgical procedures. 

The need for LAS, patients’ demographics (i.e., age, gender, body mass index [BMI] and race/ethnicity), comorbidities (i.e., American Society of Anesthesiology [ASA] physical status, history of anxiety or depression, history of smoking and alcohol disorder and history of chronic pain and opioid use), preoperative coagulation labs and postoperative clinical outcomes were obtained from electronic medical records (EMR). LAS need is typically identified at the first encounter with providers at our institution by asking the patient if they wish to have an interpreter for medical communication. This “need” is subsequently visible in the patient’s EMR. Race and ethnicity were also self-reported by patients at the time of initial registration to establish care. All patients underwent general anesthesia according to routine care in our center. The primary endpoint of the study was pain intensity (verbal numeric rating scale 0: no pain and 10: worst pain ever). Secondary endpoints included opioid use in PACU and the use of regional analgesia. 

### Statistical Analysis

A prior sample size analysis was not performed. The chi-square test or Fisher’s exact test was used to evaluate the association between two categorical variables. Wilcoxon rank-sum test was used to compare location parameters of continuous distributions between patient groups. To adjust for selection bias in this observational study, we conducted a propensity score matching (PSM) analysis. We included the following prognostic covariates in the logistic model to estimate the propensity scores: age at surgery, BMI, gender, race, ASA (1/2 vs. 3), the status of anxiety or depression and preoperative use of opioids. The Greedy 5 -> 1 digit match algorithm was used to match the baseline covariates so that the two groups (no interpreter needed vs. interpreter needed) would have similar propensity scores. A multivariable logistic regression model was fitted to estimate the effects of important covariates on the highest or average PACU pain score using 3 (mild) or 7 (severe) as the cutoff points. A *p* < 0.05 was considered statistically significant. Statistical software SAS 9.4 (SAS, Cary, NC, USA) and Splus 8.2 (TIBCO Software Inc., Palo Alto, CA, USA) was used for all the analyses.

## 3. Results

A total of 4791 patients were included from our database (Table 1). Among the patients with information, 5.4% were Asian, 7.7% were Black or African American, 13.7% were Hispanic or Latino, 1.9% were other, and 71.3% were White. A total of three hundred patients (6.26%) needed translation services. Types of primary surgeries were divided up by the surgical services that were performed; these included: colorectal, endocrine, gastric, general, gynecology, liver, melanoma, pancreas, sarcoma, thoracic and urology.

### 3.1. Pre-Matching Analysis

The pre-matching analysis demonstrated that the association between the need for translator services and the patients’ race was statically significant. Asian and Hispanic/Latino patients (n = 220, 22%) needed translation services in a higher proportion than Black/African American or White subjects (n = 77, 2%, *p* < 0.0001). A diagnosis of anxiety or depression was significantly more frequent in patients in patients needing (n = 260, 6.7%) translation services than those who were English proficient (n = 40, 4.4%, *p* = 0.011). Interestingly, the incidence of chronic opioid use was also slightly but significantly higher in patients needing (n = 100. 8.1%) translation services than those who were English proficient (n = 200.,5.6%, *p* = 0.001). Other demographic variables, including age, sex, ASA physical status, history of chronic pain, alcohol use and cigarette smoking, were not significantly different. 

In terms of length of stay, anesthesia duration, total perioperative opioid use and opioid use specifically in PACU, there were no statistically significant differences. However, PACU’s average and highest pain intensity statistically differed between English and non-English proficient patients (Table 2). The latter reported higher pain scores, although the difference was not clinically relevant (<1 unit on a scale 1/10). Similarly, pain scores on day 1 after surgery were significantly lower in English-proficient patients than those who needed translation services, but once again, the difference was not clinically relevant.

### 3.2. Post-Matching Analysis

The standardized differences for all covariates were <8% in the post-matching cohort, suggesting a substantial reduction of bias between the two groups (Table 1). After matching, a total of 590 patients (n = 295 per group) were included in the analysis. Preoperative variables were not statistically different between the patients needing LAS and the patients not needing LAS after PSM (Table 2). The rate of regional use was slightly lower but not statistically significant in patients needing LAS (47.4% vs. 52.6%; *p* = 0.16) (Table 2). The need for translation services did not impact the choice of regional block performed in either the pre- (*p* = 0.172) or post-matching (*p* = 0.9955) analysis (Table 2). Patients who needed LAS had significantly lower opioid consumption (median: 5 vs. 10; *p* = 0.021) and reported lower pain intensity during PACU stay (median: 2.3 vs. 2.6; *p* = 0.046) than subjects not requiring language translation. The difference in the percentage of mild average pain (average pain score ≥ 3) in PACU was not significant between the two groups (*p* = 0.056). Patients who needed LAS had a lower percentage of mild average pain in PACU (35% vs. 42.7%). In addition, the difference in the percentage of moderate average pain (average pain score ≥ 7) in PACU was not statistically significant between the two groups (2% vs. 2.7%; *p*-value = 0.593) (Table 2). 

A multivariable logistic regression model was fitted to estimate the effects of important covariates on the status of average PACU pain of 3 or higher. After adjusting for age, BMI, gender, race, ASA physical status, platelet count, preoperative use of opioids, regional anesthesia and status of anxiety or depression in the model, the odds of having average PACU pain of 3 or higher is 40% higher for patients not needing LAS versus patients needing LAS (odds ratio (OR) = 1.40, 95%: 0.99, 1.99). The association between needed LAS and average PACU pain of 3 or higher was not significant (*p* = 0.06). Due to the limited number of patients with average PACU pain of 7 or higher, multivariable analysis was not performed. 

## 4. Discussion

This is the first study investigating the association between LAS need and pain outcomes in patients undergoing major abdominal cancer surgery. Although the association did not reach statistical significance, we observed a 40% increase in pain (≥ 3) in patients with LAS needs compared to English-speaking patients. Proper postoperative pain assessment is integral to pain management since it can affect surgical outcomes [8]. Language barriers can impact pain management at multiple levels starting with access to adequate postoperative care [11]. The negative impact of underutilizing LAS for pain assessment has been well documented [12,13,14]. In a study of hospitalized Spanish-speaking women, Jimenez et al. reported that LAS use was associated with better pain management and patient satisfaction [12]. Amongst ambulatory patients, Latinos perceived better care when interpreters were used in pharmacies [15]. In pediatric patients, 82% (*n* = 49) of English-proficient patients and 60% of patients with LEP were treated with pain medications at the time of their worst pain [14]. Interestingly, the proportion of discordant pain reports (self-reported vs. nurse-documented) was significantly higher in children with LEP than in English-speaking patients within that study.

Time and convenience are factors often cited as barriers to effectively utilizing LAS [16,17]. This can lead to implications associated with undertreatment, as well as overtreatment of pain. Mistreatment of pain can, in turn, cause physiological and psychological dysfunction contributing to postoperative morbidity. Inadequate pain management can cause reduced quality of life, impaired sleep, impaired physical function, physiological derangements leading to chronic pain and increased healthcare costs [18]. In this study, LAS may have aided in the patient decision process regarding the acceptance of regional anesthesia. Although the need for LAS was associated with statistically significant lower pain intensity scores and a corresponding lesser opioid use than no LAS, the margin of differences, especially in pain intensity scores, may not be clinically significant. However, this still may suggest that LAS allowed for better patient-provider communication leading to more appropriate pain management. 

Our study has limitations due to its retrospective nature and confounding from unknown variables that might have influenced the decision to prescribe opioids to both groups of patients. It is a single institution study at a cancer center, thereby inclusive of only institutionally specific variables such as, but not limited to, types of surgeries and types of regional anesthesia techniques utilized. The actual use of interpreters, whether via phone, videoconferencing or in-person, could also not be retrospectively determined. Lastly, cultural differences in the psychometrics of the numeric pain scale used in our patient assessments could have influenced reported scores. 

Further research can focus on how particular modes of LAS could be best utilized in efficient and precise postoperative pain assessment. Since different institutions have different hurdles in the proper utilization of LAS, education and quality improvement initiatives at the local level can help to improve appropriate pain assessment for this vulnerable population. Our study corroborates current literature on the need for LAS to overcome communication barriers in healthcare, potentially leading to an enhanced patient-centric experience and improved clinical outcomes in patients with LEP. 

## Figures and Tables

**Table 1 jpm-13-00481-t001:** Perioperative characteristics of adults undergoing open abdominal surgery.

	Pre-Matching			Post-Matching			
	Language Assistance No (n = 4491)	Language Assistance Yes (n =300)	*p*-Value	Language Assistance No (n = 295)	Language Assistance Yes (n = 295)	Standardized Difference in %	*p*-Value
Age, median (IQR)	60 (49–69)	61 (49–68)	0.5644	59 (48–68)	60 (49–69)	7.55	
Gender, n (%) Female Male	2055 (94%) 2436 (93.5%)	132 (6%) 168 (6.5%)	0.553	138 (51.1) 157 (49.1)	132 (48.9) 163 (50.9)	1.05	
BMI, median (IQR)	27.66 (24.2–31.76)	27.03 (23.76–30.93)	0.034	26.82 (23.54 –31.02)	27.06 (23.72–31)	4.08	
ASA physical status, n (%) 1–2 3–4	245 (95.7%) 4246 (93.6%)	11 (4.3%) 289 (6.4%)	0.182	13 (54.2%) 282 (49.8%)	11 (45.8%) 284 (50.2%)	3.43	
Race/Ethnicity, n (%) Asian Black/African American Hispanic/Latino Other White	203 (79%) 367 (99.7%) 518 (79.4%) 57 (64%) 3315 (97.8%)	54 (21%) 1 (0.3%) 134 (20.6%) 32 (36%) 76 (2.2%)	<0.0001	52 (49.1) 1 (50) 132 (49.6) 34 (53.1) 76 (50)	54 (50.9) 1 (50) 134 (50.4) 30 (46.9) 76 (50)		0.980
Platelet count (cells/dL)	221 (179–275)	227.5 (179–296.5)	0.129	217 (179–273)	298 (179–298)		0.199
Anxiety or depression, n (%) No Yes	3626 (93.3%) 865 (95.6%)	260 (6.7%) 40 (4.4%)	0.011	256 (50.1) 39 (49.4)	255 (49.9) 40 (50.6)	0.99	
Chronic pain, n (%) No Yes	19 (95%)1 4472 (93.7%)	1 (5%) 299 (6.3%)	0.815	294 (50) 1 (50)	294 (50) 1 (50)		1.000
Chronic opioid use, n (%) No Yes	1131 (91.9%) 3360 (94.4%)	100 (8.1%) 200 (5.6%)	0.001	192 (49.4) 103 (51.2)	197 (50.6) 197 (50.6)	3.57	
Cigarette smoking, n (%) No Yes	10 (83.3%) 4481 (93.8%)	2 (16.7%) 298 (6.2%)	0.136	1 (33.3%) 294 (50.1%)	2 (66.7%) 293 (49.9%)		1.000
Alcohol disorder, n (%) No Yes	73 (96.1%) 4418 (93.7%)	3 (3.9%) 297 (6.3%)	0.401	287 (49.6%) 1 (33.3%)	3 (27.3%) 292 (50.4%)		0.222

ASA: American Society of Anesthesiologists, BMI: Body Mass Index, IQR: Interquartile Range.

**Table 2 jpm-13-00481-t002:** Regional anesthesia and perioperative opioid utilization, postoperative pain scores and types of surgeries performed.

	Pre-Matching			Post-Matching		
	Language Assistance No (n = 4491)	Language Assistance Yes (n =300)	*p*-Value	Language Assistance No (n = 295)	Language Assistance Yes (n = 295)	*p*-Value
Anesthesia Duration, hours (median [IQR])	6.25 (4.43–8.65)	6 (4.27–8.14)	0.11	6.38 (4.58–8.47)	5.98 (4.23–8.12)	0.0675
Regional Anesthesia No Yes	1995 (93.3%) 2496 (94.1%)	144 (6.7%) 156 (5.9%)	0.227	126 (46.8) 169 (52.6)	143 (53.2) 152 (47.4)	0.1600
Type of Regional † Epidural TAP/QL Block Other Truncal Block	1138 (93.2%) 1356 (94.9%) 2 (100%)	83 (6.8%) 73 (5.1%) . (.%)	0.172	89 (52.7%) 80 (52.6%) . (.%)	80 (47.3%) 72 (47.4%) . (.%)	0.9955
PACU opioid, MEDD (median [IQR])	5 (5–15)	5 (5–10)	0.067	10 (5–15)	5 (5–10)	0.0219
Total opioid, MEDD (median [IQR])	30 (20–50)	33 (20–50)	0.369	30 (20–45)	33 (20–50)	0.0989
Average PACU pain, (median [IQR])	2.7 (1.4–3.9)	2.3 (1–3.5)	<0.0001	2.6 (1.3–3.8)	2.3 (1.0–3.5)	0.0460
Highest PACU pain, n (%) <3 ≥3 <7 ≥7	637 (14.2%) 3850 (85.8%) 2295 (51.1%) 2192 (48.9%)	55 (18.4%) 244 (81.6%) 178 (59.5%) 121 (40.5%)	0.045 0.005	169 (57.3 %) 126 (42.7 %) 287 (97.3 %) 8 (2.7 %)	191 (65 %) 103 (35 %) 288 (98 %) 6 (2 %)	0.0560 0.5930
Average POD 1 pain (median [IQR])	2.8 (1.6–3.9)	2.4 (1.3–3.5)	<0.0001	2.8 (1.7–3.9)	2.4 (1.3–3.5)	0.0041
Types of Surgery Colorectal Endocrine Gastric General Gynecology Liver Melanoma Pancreas Sarcoma Thoracic Urology	1412 (92.3%) 140 (97.2%) 674 (95.5%) 320 (93.3%) 10 (100%) 875 (93.7%) 123 (93.2%) 332 (95.1%) 278 (93%) 23 (100%) 304 (94.7%)	118 (7.7%) 4 (2.8%) 32 (4.5%) 23 (6.7%) . (.%) 59 (6.3%) 9 (6.8%) 17 (4.9%) 21 (7%) .(.%) 17 (5.3%)	0.0885	104 (47.3%) 11 (73.3%) 46 (59%) 16 (41%) 1 (100%) 51 (47.2%) 6 (40%) 21 (55.3%) 22 (51.2%) 1 (100%) 16 (50%)	116 (52.7%) 4 (26.7%) 32 (41%) 23 (59%) . (.%) 57 (52.8%) 9 (60%) 17 (44.7%) 21 (48.8%) . (.%) 16 (50%)	0.3522

IQR: Interquartile Range, MEDD: Morphine Equivalent Daily Dose, PACU: Post Anesthesia Care Unit, POD: Postoperative Day, TAP/QL: Transversus Abdominus Plane/Quadratus Lumborum. †Information on specific types of blocks was not available for all patients who received regional analgesic techniques.

## Data Availability

Data is available on request due to institutional restrictions. The data presented in this study are available on request from the corresponding author.

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
