# Peer review of "The Impact of the Need for Language Assistance Services on the Use of Regional Anesthesia, Postoperative Pain Scores and Opioid Administration in Surgical Oncology Patients"

_jpm, 2023, doi:10.3390/jpm13030481_

Round 1
Reviewer 1 Report
The study is very interesting from a practical point of view. The problem of communication affects many communities especially with so much migration.
It is necessary in the discussion section to expand the comparison of other papers. Although this is a unique study, but in my opinion it is worth expanding the comparison with papers on patients with impaired communication with the environment, perhaps also in pediatric patients.
The problem of communication with the environment and its impact on pain management has been described in studies and it is worth addressing.
Please expand the limitations.
Please increase the number of papers cited and refer to them in the text this should increase the importance and relevance of the article.
Author Response
Thank you for your comments. We agree that this study is unique and therefore is difficult to directly compare with other papers. As suggested, we have expanded the discussion with a relevant and very recent citation focusing on how language barriers can impact opioid prescription use after TKA. We have also expanded the limitations section as recommended.
Reviewer 2 Report
Excellent idea regarding that postoperative pain control has it advantages in early discharge and especially in the morbidity control. Its easy to understand that if you cannot make appropriate verbal contact with the patient that you can not make sometimes proper decisions. I liked the idea and the concept behing this paper.
The concept is promising but the statistics regarding better pain control is lacking especially those with LAS - also it would be wise to point out which type of surgery was done because abdominal oncology surgery do differ between them and make different pain scores postoperatively (also duration of the surgery and perioperative use of opioids/regional blocks) and definitely which type of regional block (anesthesia) was performed. VAS analogue scale was made to make sure that patients just need to point to the number which corresponds there pain score.
Most of the patients did not have precipitating factors such as anxiety/depression, chronic pain syndromes or were using preoperatively pain medications or adjuvants which is good especially in acute postoperative control - maybe because of that same language barrier that they did not have the proper care in the first place.
Author Response
Thank you for your comments. We have updated Table 2 to include types of surgeries, types of regional anesthesia, anesthesia duration (a correlate for surgical duration), and total opioids utilized. Additionally, we commented on these variables in the discussion section.
Reviewer 3 Report
Although this project has the potential to improve quality of care in the PACU, or the surgical floor, it lacks generalized viewpoints.
As a retrospective study, a lot of data are missing - why did the authors decide to only show the propensity score matching (PSM) analysis? Since cultural/ psychometric information has been lost to time and exact LAS use could not be derived from the records, how is the PMS equivalent to data balancing?
It is important to accumulate all the information from this retrospective study. Keep the PMS analysis, since it does further the current knowledge on the subject - albeit minimally as there is no statistical significance. However, please add analyses obtained from the cohort you chose to study: Surgical Oncology Patients needing LAS and not needing LAS. For instance, did gender have an impact on the need for LAS? Did gender+needing LAS differ betweeen ethnicities? These are mere examples, please reconsider the data analysis, especially since your results did not present statistically important conclusions. There is so much to be learnt by this cohort and the "balanced data" from the PMS is highly derivative.
Also, in Table 2 / last line there is an important difference in the first post-op day pain scory, since pain intensity is your main objective, why was this not mentioned/commented upon? Is the strict threshold of VAS3 preventing the authors from commenting on this statistically significant finding?
The "track changes" text has a few minor corrections also.

Author Response
Thank you for your comments. We have provided the analysis before PSM. Gender was used in matching and we have mentioned this in the results section. We have also commented in the results section that although PACU's average and highest pain intensity statistically differed between English and non-English proficient patients, the difference was not clinically relevant. We did the same for pain scores on POD1 after surgery.
Round 2
Reviewer 2 Report
Good job.
Reviewer 3 Report
Impressive changes - accepted.